# SMOOTHING SLOT ATTENTION ITERATIONS AND RE-CURRENCES

## ABSTRACT

Slot Attention (SA) and its variants lie at the heart of mainstream Object-Centric Learning (OCL). Objects in an image can be aggregated into corresponding slot vectors, by *iteratively* refining cold-start query vectors, typically three times, via SA on image features. For video, this aggregation is *recurrently* shared across frames, with queries cold-started on the first frame while transitioned from the previous frame's slots on non-first frames. However, cold-start queries lack sample-specific cues thus hindering precise aggregation on the image or video's first frame; Also, non-first frames' queries are already sample-specific thus requiring aggregation transforms different from the first frame. We address these issues for the first time with our *SmoothSA*: (1) To smooth SA iterations on the image or video's first frame, we *preheat* the cold-start queries with rich information of input features, via a tiny module self-distilled inside OCL; (2) To smooth SA recurrences across all video frames, we *differentiate* the homogeneous transforms on the first and non-first frames, by using full and single iterations respectively. Comprehensive experiments on object discovery, recognition and downstream benchmarks validate our method's effectiveness. Further analyses illuminate how our method smooths SA iterations and recurrences. Our source code and training logs are provided in the supplement.

## 1 INTRODUCTION

Object-Centric Learning (OCL) (Locatello et al., 2020) aims to represent objects in a visual scene as distinct vectors, with the background as another vector. Ideally, this yields a structured compact representation that outperforms popular dense feature maps in advanced vision tasks. In dynamics modeling, evolving these object-level slots over time captures more accurate object interactions (Villar-Corrales & Behnke, 2025). For visual reasoning, their concise form allows more explicit object relationship modeling, slashing the search space and computation load (Ding et al., 2021). In visual prediction, disentangling objects facilitates more compositional generation of future frames (Villar-Corrales et al., 2023).

Powered by Slot Attention (SA) (Locatello et al., 2020), modern OCL methods have significantly improved and can now scale to real-world complex images and videos. SA is essentially a form of iterative cross attention, where query vectors compete to aggregate their corresponding object information, discovering objects as segmentation masks and representing them as slot vectors (Locatello et al., 2020).

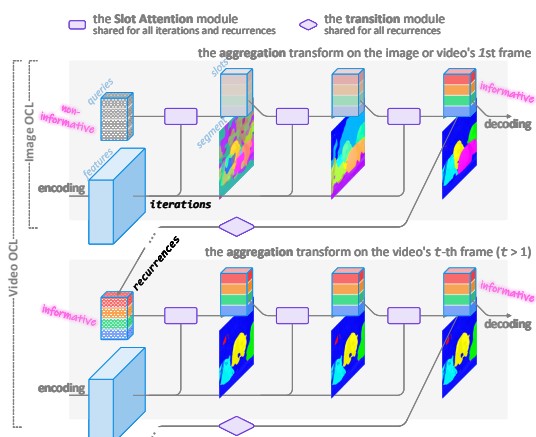

Figure 1: Image Object-Centric Learning (OCL) is realized via Slot Attention (SA) *iterations* on the image, while video OCL is via SA *recurrences* across video frames. In SA iterations on the image or video's first frame, the **cold-start queries** lack information for accurate aggregation; In SA recurrences across video's first and non-first frames, the **homogeneous transforms**, i.e., the fixed three SA iterations, cannot jointly adapt to the first and non-first queries, which have a significant information gap.

The model is trained by minimizing reconstruction loss based on the slots, requiring no external supervision. Specifically, for image, the queries are usually cold-start and sampled from multiple Gaussian distributions fitted to the entire dataset (Jia et al., 2023). Such queries contain no information about any specific sample, thus to obtain slots by refining queries using SA on image features, typically three iterations are necessary. For video, such aggregation occurs recurrently across all frames in a shared way, where queries for the first frame are the same as in the image case while queries for non-first frames are transitioned from the previous frame's slots (Singh et al., 2022b). Unlike the first frame's queries, non-first frames' queries are already quite sample-specific. But the aggregation transforms are identical or homogeneous across all frames.

To the best of our knowledge, all works on SA and its variants confront these facts but have not acknowledged the implied issues, as shown in Figure 1: (*i1*) *Query cold-start* in SA iterations. For an image or video's first frame, the cold-start queries lack scene-specific information. Although three SA iterations can gradually refine these non-informative queries into useful slots, such aggregation would not work as good as that with informative queries. (*i2*) *Transform homogeneity* in SA recurrences. For video frames, the first frame's queries are cold-start while non-first frames' are much more informative. These differing conditions impose different requirements on the aggregation transforms, thus such homogeneous transforms would not work as good as those adapted to informative-different queries.

Our solution is simple yet effective. We propose *SmoothSA*, which smooths SA iterations on the image or video's first frame by preheating the queries, and smooths SA recurrences across video's first and non-first frames by differentiating the transforms: (*s1*) A tiny module *preheats* the cold-start queries using rich information from input features. It is trained by predicting current slots through self-distillation within the OCL model. (*s2*) Different aggregation transforms handle video's first and non-first frames respectively. This is realized by simply employing three SA iterations on the first frame while only one on each non-first frame.

Briefly, our contributions are: (*c1*) for the first time addressing the query cold-start issue in SA iterations on the image and video's first frame; (*c2*) for the first time addressing the transform homogeneity issue in SA recurrences across the first and non-first frames; (*c3*) new state-of-the-art on both image and video OCL benchmarks; (*c4*) consistent performance boosts on downstream advanced vision tasks.

## 2 RELATED WORK

As SA is a kind of cross attention that depends on queries to aggregate information from visual features, we review works from perspectives of aggregation and queries.

**Slot Attention on images and videos**. The seminal work on the aggregation module SA (Locatello et al., 2020) proposes refining the initial randomly initialized queries into object-centric slots via typically three iterations of the same SA module on image features. Then, all image OCL methods including (Singh et al., 2022a; Seitzer et al., 2023; Wu et al., 2023b; Jiang et al., 2023; Kakogeorgiou et al., 2024; Zhao et al., 2025b;c;d;e) adopt this iterative design. The pioneering work STEVE (Singh et al., 2022b) extends SA to videos by conducting standard image OCL on each frame, using randomly initialized queries for the first frame while using recurrently predicted queries from previous slots for non-first frames. After, all video OCL methods including SAVi (Kipf et al., 2022), SAVi++ (Elsayed et al., 2022), SOLV (Aydemir et al., 2023), VideoSAUR (Zadaianchuk et al., 2024), SlotContrast (Manasyan et al., 2025), STATM (Li et al., 2025b), SlotPi (Li et al., 2025a) and RandSF.Q (Zhao et al., 2025a) adopt such recurrent design. Now that SA is the core module of mainstream OCL methods for images or videos, all methods face but never acknowledge two issues described above. Our method is the first to address these issues directly.

**Query initialization for Slot Attention iterations**. For images, the initial queries serve as the starting point for aggregation based on SA iterations. The principal contradiction is that no object cues are available before aggregation. SA (Locatello et al., 2020) initializes queries by drawing multiple samples from a global Gaussian distribution, which is learned on the entire dataset and embeds global cues for object discovery. BO-QSA (Jia et al., 2023) proposes learning multiple Gaussian distributions so that more distinct cues are embedded into initial queries, thus enabling better aggregation. However, the queries are still cold-start. MetaSlot (Liu et al., 2025) takes two

steps: firstly initializing queries from multiple Gaussians for draft aggregation iterations, and then replacing the draft slots with object embeddings from a large codebook (Van Den Oord et al., 2017) for additional aggregation iterations. This mitigates the iterative query cold-start effectively, but still relies on cold-start queries. We directly address such iterative query cold-start issue.

**Query prediction for Slot Attention recurrences**. For video's first frame, the queries can be obtained in the same way as in the image case, or by transforming cues like object bounding boxes in SAVi (Kipf et al., 2022) and SAVi++ (Elsayed et al., 2022), albeit at the cost of extra expensive annotations. For non-first frames, the queries are predicted from the previous frame's slots. STEVE (Singh et al., 2022b) and most other OCL methods use a Transformer encoder block for such recurrent prediction. STATM (Li et al., 2025b) and SlotPi (Li et al., 2025a) employ auto-regressive Transformer encoder variants for the same purpose. The most recent work RandSF.Q (Zhao et al., 2025a) incorporates the next frame's feature for more informative query prediction, and uses random slot-feature pairs for explicit query prediction learning, significantly boosting OCL performance on videos. However, improving query prediction alone will never reach the core issue, recurrent transform discrepancy. We directly address this recurrent transform homogeneity issue.

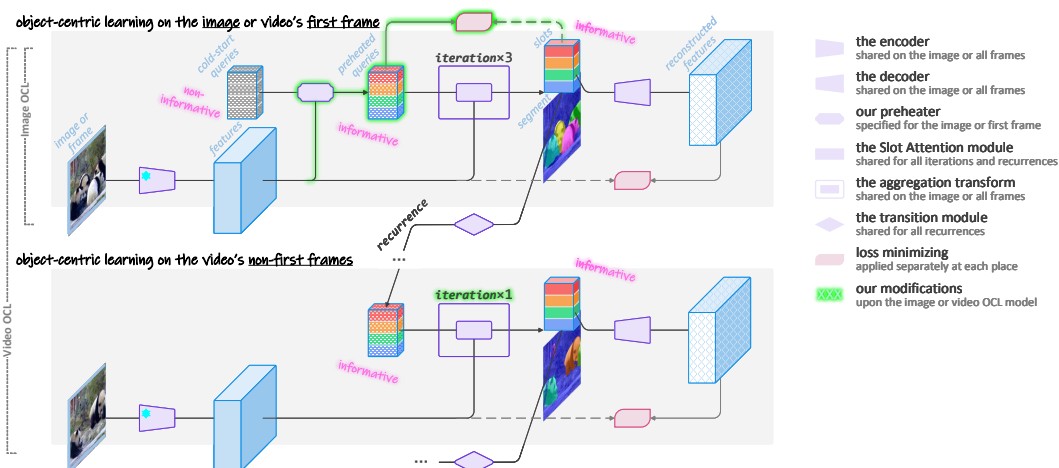

Figure 2: The overall model and our modifications. (*upper*) For image OCL, we preheat the cold-start queries to be informative so as to smooth SA iterations on the image (or video's first frame). Our preheater is a tiny module trained to predict vectors approximating the slots as the preheated queries from the cold-start queries and image features. (*upper + lower*) For video OCL, we differentiate the homogeneous transforms to adapt to the first and non-first queries, non-informative and informative respectively, to smooth SA recurrences across all frames. This is achieved by using three SA iterations on the first frame and one on non-first frames.

## 3 PROPOSED METHOD

Mainstream image or video OCL methods confront two issues: the query cold-start in SA iterations on the image or video's first frame, and the transform homogeneity in SA recurrences across video's first and non-first frames. We address these issues for the first time with our *SmoothSA*, by preheating queries to smooth SA iterations and differentiating transforms to smooth SA recurrences.

### 3.1 SLOT ATTENTION ITERATION AND RECURRENCE

Mainstream OCL methods mainly take the encoder-aggregator-decoder model design (Zhao et al., 2025d): The encoder encodes the image or video frames into features, the aggregator aggregates features into slots, and the decoder decodes slots into the reconstruction of the input in some form as the source of supervision. The aggregator, which is based on Slot Attention (SA) (Locatello et al., 2020) or its variants, is the core of OCL, so let us focus on it.

**SA iterations on the image or video's first frame**. An SA-based aggregator $\phi_a$ takes cold-start vectors $Q_1 \in \mathbb{R}^{n \times c}$ as the query, and input features $F_1 \in \mathbb{R}^{h \times w \times c}$ as the key and value. $\phi_a$ is

applied on the query, key and value typically three times, to refine the query iteratively into object-level feature vectors $S_1 \in \mathbb{R}^{n \times c}$, i.e., slots, as the sparse representation of the visual scene:

$$Q_1 = \phi_n(C) \tag{1}$$

$$S_1, M_1 = \Phi_a(Q_1, F_1) \tag{2}$$

where the aggregation transform $\Phi_a$ can be expanded into:

$$S_1^{(0)} := Q_1 \tag{2a}$$

$$S_1^{(i)}, M_1^{(i)} = \phi_a(S_1^{(i-1)}, F_1) \quad i = 1, 2, 3 \tag{2b}$$

$$S_1, M_1 := S_1^{(3)}, M_1^{(3)} \tag{2c}$$

In Equation (1), if cues $C$ are $n$ slots to use, then the initializer $\phi_n$ samples $n$ vectors as the queries $Q_1$ from its trainable Gaussian distribution(s) (Locatello et al., 2020; Jia et al., 2023); If cues $C$ are the bounding boxes of objects in the video's first frame, then the initializer $\phi_n$ projects cues $C$ into the queries $Q_1$ (Kipf et al., 2022; Elsayed et al., 2022). In whichever case, queries $Q_1$ lack sample-specific information, namely, being cold-start.

Considering that $F_1$ is the high-quality feature of the image or video's first frame, typically extracted by vision foundation model DINO2 (Oquab et al., 2023), the quality of the transform $\Phi_a$ is decided by queries $Q_1$. Therefore, if we could preheat the cold-start queries $Q_1$ to be more informative, the aggregation transform $\Phi_a$ on the image or video's first frame would perform better.

**SA recurrences across video's first and non-first frames**. The transform $\Phi_a$ based on SA iterations is shared across all frames recurrently. Namely, the transform $\Phi_a$ happens across both first and non-first frames, where the former is identical to the image case formulated in Equations (1) and (2). For the latter, queries $Q_t$ are recurrently transitioned from previous frame's slots $S_{t-1}$:

$$Q_t = \phi_r(S_{t-1}) \quad t \geq 2 \tag{3}$$

$$S_t, M_t = \Phi_a'(Q_t, F_t) \tag{4}$$

where the aggregation transform $\Phi_a'$ can be expanded into:

$$S_t^{(0)} := Q_t \tag{4a}$$

$$S_t^{(i)}, M_t^{(i)} = \phi_a(S_t^{(i-1)}, F_t) \quad i = 1, 2, 3 \tag{4b}$$

$$S_t, M_t := S_t^{(3)}, M_t^{(3)} \tag{4c}$$

In Equation (3), the transitioner $\phi_r$ takes previous frame's slots $S_{t-1}$ as input and predicts current queries $Q_t$. Considering that $S_{t-1}$ is the information-intensive representation of the previous frame and that the transitioner $\phi_r$ learns knowledge of transition dynamics (Singh et al., 2022b), current queries $Q_t$ is actually informative to current frame. This is different from the first frame queries $Q_1$, which is cold-start and thus non-informative.

The non-first frames' transform shares exactly the same SA module from the first transform with the same number of SA iterations, i.e., $\Phi_a' \equiv \Phi_a$. On the other hand, the information gap between the first queries $Q_1$ and non-first queries $Q_t$ imposes different requirements on these transforms. Therefore, if we could differentiate the homogeneous transforms $\Phi_a$ and $\Phi_a'$ for the first and non-first frames respectively, the aggregation across the first and non-first frames would be better.

### 3.2 PREHEATING COLD-START QUERIES

To overcome the query cold-start issue and smooth SA iterations on the image or video's first frame, we preheat the cold-start queries with rich information from input features. A tiny module is trained via self-distillation inside the OCL model, to predict vectors that approximate the aggregated slots as the preheated queries, from the cold-start queries conditioned on input features.

Our chain-of-thought is as follows: (*i*) Informative slots can be aggregated by iteratively refining non-informative queries; (*ii*) More informative queries contribute to better slots aggregation; (*iii*)

How to preheat the queries to be more informative? (*iv*) Aligning the preheated queries with the aggregated slots, which are quite informative.

Firstly, we insert this between Equations (1) and (2):

$$\boldsymbol{Q}_1^* = \phi_{\mathrm{p}}(\boldsymbol{Q}_1, \boldsymbol{F}_1) \tag{5}$$

where the preheater $\phi_{\mathrm{p}}$ is parameterized as a single Transformer decoder block (Vaswani et al., 2017), whose self-attention and cross-attention are switched. This is because exchanging information among non-informative queries firstly is meaningless. Please refer to Table 5 ablation studies for why not using an extra SA module as the preheater, and for why switching the self-attention and cross-attention.

Secondly, we replace Equation (2a) with:

$$\boldsymbol{S}_1^{(0)} := \mathrm{sg}(\boldsymbol{Q}_1^*) \tag{6}$$

where $\mathrm{sg}(\cdot)$ is stopping gradient. Stopping gradient flow from the SA module $\phi_{\mathrm{a}}$ to the preheated queries $\boldsymbol{Q}_1^*$ disentangles the training of $\phi_{\mathrm{a}}$ and $\phi_{\mathrm{p}}$. Please refer to Table 5 ablation studies for why stopping gradient flow on the preheated queries.

Lastly, to obtain the preheating ability, we train our preheater $\phi_{\mathrm{p}}$ with the following objective:

$$\arg \min_{\boldsymbol{C}, \phi_{\mathrm{n}}, \phi_{\mathrm{p}}} \mathrm{MSE}(\boldsymbol{Q}_1^*, \mathrm{sg}(\boldsymbol{S}_1)) \tag{7}$$

where the MSE loss is combined with the original OCL loss(es). To ensure the sufficient training of $\phi_{\mathrm{p}}$, we can use a relatively large coefficient for it. Please refer to Table 5 ablation studies for what weight to set for such preheating loss.

**Comment 1**. Our preheater is trained with OCL intermediate results as the ground-truth, without any external supervision, forming rigid self-distillation. This is also bootstrap, as good slots $\boldsymbol{S}_1$ leads to better preheated queries $\boldsymbol{Q}_1^*$, and in turn better $\boldsymbol{Q}_1^*$ leads to better $\boldsymbol{S}_1$.

**Comment 2**. Our preheater is similar to the SA module, without an heavy RNN module. Thus our preheater introduces approximately less than 1/3 more computation overhead in the aggregation, which is negligible considering the heavy computation of encoding and decoding.

### 3.3 DIFFERENTIATING HOMOGENEOUS TRANSFORMS

To overcome the transform homogeneity issue and smooth SA recurrences across the video's first and non-first frames, we differentiate the homogeneous transforms for the first and non-first frames respectively. For the different transform requirements due to the gap between the first cold-start queries and non-first informative queries, full and single SA iterations are used respectively.

Our chain-of-though is: (*i*) First frame queries are non-informative, thus three SA iterations are needed to refine the queries into good slots; (*ii*) Non-first frame queries are already informative, thus a single SA iteration is enough.

As mentioned above, the first-frame transform $\boldsymbol{\Phi}_{\mathrm{a}}$ and non-first frame transforms $\boldsymbol{\Phi}_{\mathrm{a}}'$ are identical in all existing methods but should be different. There are two ways to differentiate them: (1) use separate SA parameters for $\boldsymbol{\Phi}_{\mathrm{a}}$ and $\boldsymbol{\Phi}_{\mathrm{a}}'$; (2) use different number of iterations for $\boldsymbol{\Phi}_{\mathrm{a}}$ and $\boldsymbol{\Phi}_{\mathrm{a}}'$. We choose the second solution. This is because $\boldsymbol{\Phi}_{\mathrm{a}}$ and $\boldsymbol{\Phi}_{\mathrm{a}}'$ should learn the general aggregation capability in each SA iteration and sharing enforces this. Please refer to Table 5 ablation studies for what numbers of iterations for first and non-first transforms to set.

We simply reduce the number of SA iterations in non-first frame transforms $\boldsymbol{\Phi}_{\mathrm{a}}'$ to once, while always use three SA iterations in the first frame transform $\boldsymbol{\Phi}_{\mathrm{a}}$. Namely, we keep Equations (2b) and (2c) unchanged, while replacing Equations (4b) and (4c) with:

$$\boldsymbol{S}_t^{(i)}, \boldsymbol{M}_t^{(i)} = \phi_{\mathrm{a}}(\boldsymbol{S}_t^{(i-1)}, \boldsymbol{F}_t) \quad i = 1 \tag{8b}$$

$$\boldsymbol{S}_t, \boldsymbol{M}_t := \boldsymbol{S}_t^{(1)}, \boldsymbol{M}_t^{(1)} \tag{8c}$$

For conditional SA like in SAVi (Kipf et al., 2022) and SAVi++ (Elsayed et al., 2022), they use homogeneous aggregation transforms, consisting of one single SA iteration for all frames. But we

still use three SA iterations on the first frame and one on non-first frames. They believe that objects' bounding boxes as query initialization is informative enough. But in fact, they still carry little object information, except the spatial information. Thus more iterations on the first frame is still necessary. Their ablation study leads them to believe that one iteration is better than more just because they were not aware of such recurrent transform homogeneity issue. Please refer to Table 5 ablation studies for what numbers of iterations for first and non-first transforms to set.

**Comment 3**. Our differentiated transforms have 2/3 less computation overhead in aggregation on non-first frames, which is negligible considering the heavy encoding and decoding.

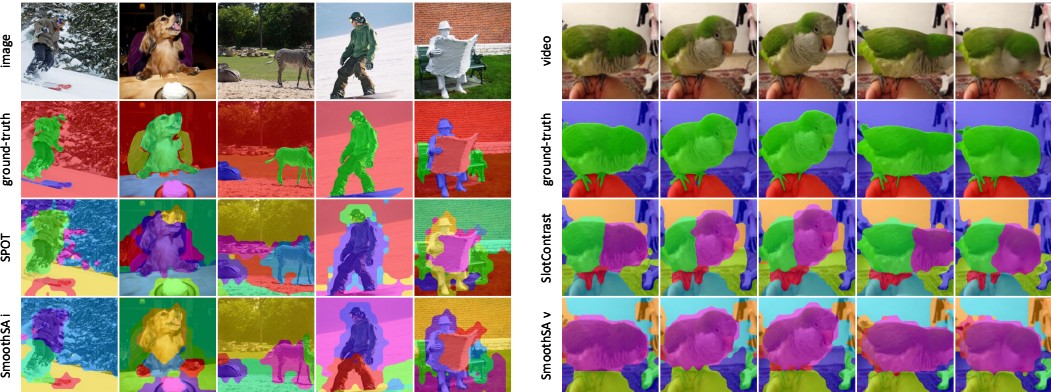

Figure 3: Qualitative results of our SmoothSA on images (*left*) and videos (*right*), compared with SotA methods SPOT and SlotContrast respectively.

## 4 EXPERIMENT

We conduct experiments on object discovery along with downstream tasks, object recognition and visual question answering, to evaluate our slots representation quality, with three random seeds.

### 4.1 INSTANTIATING SMOOTHSA

As shown in Figure 2, our OCL model with SmoothSA is based on DIAS$^i$ (Zhao et al., 2025e) for images and RandSF.Q (Zhao et al., 2025a) for videos, respectively. These two state-of-the-art (SotA) methods share identical designs except techniques specific to image and video. For image OCL, we remove slots pruning tricks from DIAS$^i$, and then replace its SA variant with our SmoothSA. For video OCL, we use RandSF.Q as it is, and then replace its SA with our SmoothSA. Thus we have models SmoothSA$^i$ and SmoothSA$^v$, where $i$ is image and $v$ is video.

Note that for conditional video OCL like SAVi (Kipf et al., 2022) and SAVi++ (Elsayed et al., 2022), the authors always use one SA iteration on all frames. But whether it is conditional or not, we always use three SA iterations on the first frame while one iteration on non-first frames.

| | ClevrTex #slot=11 | | | | COCO #slot=7 | | | | VOC #slot=6 | | | |
|---|---|---|---|---|---|---|---|---|---|---|---|---|
| | ARI | ARI$_{fg}$ | mBO | mIoU | ARI | ARI$_{fg}$ | mBO | mIoU | ARI | ARI$_{fg}$ | mBO | mIoU |
| SLATE | 17.4$_{\pm2.9}$ | 87.4$_{\pm1.7}$ | 44.5$_{\pm2.2}$ | 43.3$_{\pm2.4}$ | 17.5$_{\pm0.6}$ | 28.8$_{\pm0.3}$ | 26.8$_{\pm0.3}$ | 25.4$_{\pm0.3}$ | 18.6$_{\pm0.1}$ | 26.2$_{\pm0.8}$ | 37.2$_{\pm0.5}$ | 36.1$_{\pm0.4}$ |
| DINOSAUR | 50.7$_{\pm24.1}$ | 89.4$_{\pm0.3}$ | 53.3$_{\pm5.0}$ | 52.8$_{\pm5.2}$ | 18.2$_{\pm1.0}$ | 37.0$_{\pm1.2}$ | 28.3$_{\pm0.5}$ | 26.9$_{\pm0.5}$ | 21.5$_{\pm0.7}$ | 36.2$_{\pm1.3}$ | 40.6$_{\pm0.6}$ | 39.7$_{\pm0.6}$ |
| SlotDiffusion | 66.1$_{\pm1.3}$ | 82.7$_{\pm1.6}$ | 54.3$_{\pm0.5}$ | 53.4$_{\pm0.8}$ | 17.7$_{\pm0.5}$ | 29.0$_{\pm0.1}$ | 27.0$_{\pm0.4}$ | 25.6$_{\pm0.4}$ | 17.0$_{\pm1.2}$ | 21.7$_{\pm1.8}$ | 35.2$_{\pm0.9}$ | 34.0$_{\pm1.0}$ |
| SPOT | 25.6$_{\pm1.4}$ | 77.1$_{\pm0.7}$ | 48.3$_{\pm0.5}$ | 46.4$_{\pm0.6}$ | 20.0$_{\pm0.5}$ | 40.0$_{\pm0.7}$ | 30.2$_{\pm0.3}$ | 28.6$_{\pm0.3}$ | 20.3$_{\pm0.7}$ | 33.5$_{\pm1.1}$ | 40.1$_{\pm0.5}$ | 38.7$_{\pm0.7}$ |
| DIAS$^i$ | 80.9$_{\pm0.3}$ | 79.1$_{\pm0.3}$ | 63.3$_{\pm0.0}$ | 61.9$_{\pm0.0}$ | 22.0$_{\pm0.2}$ | 41.4$_{\pm0.2}$ | 31.1$_{\pm0.1}$ | 29.7$_{\pm0.1}$ | 26.6$_{\pm1.0}$ | 33.7$_{\pm1.5}$ | 43.3$_{\pm0.3}$ | 42.4$_{\pm0.3}$ |
| SmoothSA$^i$ | 76.8$_{\pm1.4}$ | 80.8$_{\pm1.6}$ | 60.0$_{\pm1.8}$ | 58.1$_{\pm2.2}$ | 26.2$_{\pm0.8}$ | 42.1$_{\pm0.7}$ | 33.2$_{\pm0.4}$ | 31.7$_{\pm0.4}$ | 30.6$_{\pm0.6}$ | 34.3$_{\pm0.5}$ | 45.3$_{\pm0.5}$ | 44.1$_{\pm0.6}$ |

Table 1: Object discovery on images. Input resolution is 224×224; DINO2 ViT-S/14 is for encoding.

## 4.2 Object Discovery

In mainstream OCL methods, attention maps of the slots are binarized as the byproduct object segmentation, i.e., discovering objects. This intuitively reflects slots' representation quality.

On image datasets ClevrTex[1], COCO[2] and VOC[3], we compare our SmoothSA$^i$ with baselines SLATE (Singh et al., 2022a), DINOSAUR (Seitzer et al., 2023), SlotDiffusion (Wu et al., 2023b), SPOT (Kakogeorgiou et al., 2024) (no distillation and finetuning tricks) and DIAS (Zhao et al., 2025e) (no slot pruning). On video dataset YouTube Video Instance Segmentation[4] (YTVIS) the high-quality version[5], we compare our SmoothSA$^v$ with baselines STEVE (Singh et al., 2022b), VideoSAUR (Zadaianchuk et al., 2024), SlotContrast (Manasyan et al., 2025) and RandSF.Q (Zhao et al., 2025a). The performance metrics are ARI[6], ARI$_{fg}$ (foreground), mBO (Uijlings et al., 2013) and mIoU[7]. ARI score is calculated with the segmentation area as the weight, thus ARI mainly reflects how well the background is segmented while ARI$_{fg}$ reflects how well large objects are segmented. mBO shows how objects that are best overlapped with the ground-truth are segmented. mIoU is the most strick metric. Note that, unless otherwise specified, we use image ARI, ARI$_{fg}$, mBO and mIoU for object discovery on images, while using video ones for object discovery on videos. Also note that on dataset YTVIS, we use video clip length 5 for training while 20 for evaluation.

|  | ARI | ARI$_{fg}$ | mBO | mIoU |
|---|---|---|---|---|
| YTVIS #slot=7, #step=20 | | | | |
| VideoSAUR | $34.6_{\pm0.5}$ | $48.6_{\pm0.7}$ | $31.4_{\pm0.3}$ | $31.2_{\pm0.3}$ |
| SlotContrast | $38.7_{\pm0.9}$ | $48.9_{\pm0.9}$ | $35.0_{\pm0.3}$ | $34.9_{\pm0.3}$ |
| DIAS$^v$ | $33.6_{\pm0.4}$ | $49.3_{\pm0.7}$ | $36.1_{\pm1.4}$ | $35.2_{\pm0.8}$ |
| RandSF.Q | $42.0_{\pm0.3}$ | $59.4_{\pm1.4}$ | $39.8_{\pm0.3}$ | $39.4_{\pm0.3}$ |
| SmoothSA$^v$ | $44.1_{\pm1.8}$ | $61.5_{\pm3.2}$ | $41.1_{\pm1.4}$ | $40.6_{\pm1.4}$ |

Table 2: Object discovery on videos. Input resolution is $224\times224$; DINO2 ViT-S/14 is for encoding.

As shown in Table 1, on synthetic dataset ClevrTex, our SmoothSA$^i$ is as competitive as the latest SotA DIAS$^i$ and significantly better than former SotA SPOT in all metrics. On real-world dataset COCO, our SmoothSA$^i$ is consistently better than DIAS$^i$ in all metrics, 4+ points in ARI. On real-world dataset VOC, our method pushes the ARI value forward by 4 points. Our method achieves overall new SotA in ARI, mBO and mIoU, except relative limited performance boosts in ARI$_{fg}$.

As shown in Table 2, on real-world video dataset YTVIS, our SmoothSA$^v$ defeats all baselines by a large margin, even including the latest super SotA method RandSF.Q, which has already pushed the older SotA performance significantly forward by up to 10 points.

## 4.3 Object Recognition

Besides the byproduct segmentation, recognizing the discovered objects' attributes like class and bounding box from the slots can directly reflect the object-centric representation quality.

On real-world image dataset COCO, we compare our SmoothSA$^i$ with baseline SPOT (Kakogeorgiou et al., 2024). On real-world video dataset YTVIS, we compare our SmoothSA$^v$ with baseline SlotContrast (Manasyan et al., 2025). We follow the routine of (Seitzer et al., 2023): firstly convert all images into slots representation, with some threshold filtering; then train a two-layer MLP model to classify and regress the matched object's class label and bounding box coordinates in a supervised way. We use top1

|  |  | class top1↑ | bbox R2↑ |
|---|---|---|---|
|  |  | COCO #slot=7 | |
| SPOT | + MLP | $0.67_{\pm0.0}$ | $0.62_{\pm0.1}$ |
| SmoothSA$^i$ | + MLP | $0.73_{\pm0.0}$ | $0.64_{\pm0.1}$ |
|  |  | YTVIS #slot=7, #step=20 | |
| SlotContrast | + MLP | $0.40_{\pm0.1}$ | $0.53_{\pm0.1}$ |
| SmoothSA$^v$ | + MLP | $0.50_{\pm0.0}$ | $0.62_{\pm0.0}$ |

Table 3: Object recognition on image dataset COCO and video dataset YTVIS.

---

[1] https://www.robots.ox.ac.uk/˜vgg/data/clevrtex

[2] https://cocodataset.org

[3] http://host.robots.ox.ac.uk/pascal/VOC

[4] https://youtube-vos.org/dataset/vis

[5] https://github.com/SysCV/vmt?tab=readme-ov-file#hq-ytvis-high-quality-video-instance-segmentation-dataset

[6] https://scikit-learn.org/stable/modules/generated/sklearn.metrics.adjusted_rand_score.html

[7] https://scikit-learn.org/stable/modules/generated/sklearn.metrics.jaccard_score.html

accuracy[8] to measure the classification performance,
and R2 score[9] to measure the regression performance.

As shown in Table 3, the object recognition accuracy on both real-world complex images and videos are improved a lot by using our method as the slots representation extractor, compared with that using baseline methods. This demonstrates the high quality of our slots representation.

### 4.4 VISUAL QUESTION ANSWERING

In visual question answering (VQA) tasks, the visual modality slots are combined with language modality words embeddings together, testing the representation quality and versatility further.

For VQA on images, we compare our SmoothSA$^i$ plus multi-modal reasoning model Aloe (Ding et al., 2021) with baseline SPOT plus Aloe on real-world complex image dataset GQA[10]. For VQA on videos, we compare our SmoothSA$^v$ plus Aloe with baseline Slot-Contrast plus Aloe on synthetic video dataset CLEVRER[11]. Please note that for the image dataset, we use Aloe as it is while on the video dataset we introduce temporal embedding scheme from (Wu et al., 2023a). For the upstream OCL models, we firstly pretrain them on corresponding datasets and freeze

|  |  | GQA #slot=7 | |
|---|---|---|---|
|  |  | accuracy % | |
| SPOT | + Aloe | 52.3$_{\pm 2.8}$ | |
| SmoothSA$^i$ | + Aloe | 56.7$_{\pm 1.9}$ | |
|  |  | CLEVRER #slot=7 | |
|  |  | per option % | per question % |
| SlotContrast | + Aloe | 97.2$_{\pm 1.1}$ | 95.6$_{\pm 0.9}$ |
| SmoothSA$^v$ | + Aloe | 98.7$_{\pm 0.4}$ | 96.9$_{\pm 0.6}$ |

Table 4: Visual question answering on image dataset GQA and video dataset CLEVRER.

them to represent samples as slots. These visual input along with textual inputs representing questions are fed into the Aloe model together, appended with a classification token. The output is obtained by projecting the transformed classification token into logits of all possible class labels, i.e., answers.

As shown in table 4, using our method as the upstream model improves the image VQA performance on dataset GQA by 4+ points. As for video VQA on CLEVRER, using our method as the upstream boosts the performance too, whether measured by per option accuracy or per question accuracy.

## 5 DISCUSSION

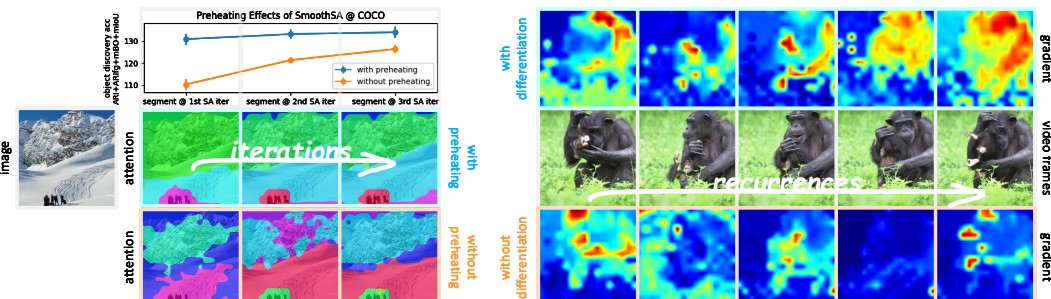

Figure 4: (*left*, *middle row*) Using query preheating, good segmentation can be obtained at the beginning of SA iterations, with even better segmentation at the end. (*right*, *top row*) Using transform differentiation, balanced gradient signals can be obtained across SA recurrences, even showing good object contours.

**Query preheating smooths SA iterations**

The segmentation accuracies generally increase along with the SA iterations, so we expect that our query preheating provides better initial queries and the accuracies increase faster. Please refer to Sec-

---

[8]https://scikit-learn.org/stable/modules/generated/sklearn.metrics.accuracy_score.html

[9]https://scikit-learn.org/stable/modules/generated/sklearn.metrics.r2_score.html

[10]https://cs.stanford.edu/people/dorarad/gqa

[11]http://clevrer.csail.mit.edu

tion A.4 for our mathematical analysis. To statistically analyze this, we training our SmoothSA$^i$ with and without query preheating on COCO, and count their respective object discovery/segmentation accuracy at each of the three SA iterations. In practice, those four metrics mentioned above, i.e., ARI, $ARI_{fg}$, mBO and mIoU, mostly show cluttered tendencies in iterations, thus we sum them together as a more readable metric.

As shown in Figure 4 (*left*) the top row, on the whole dataset, segmentation accuracies with or without query preheating at three SA iterations increase steadily. But using query preheating obviously speeds it up and leads to better final accuracy than not. As shown in Figure 4 (*left*) the lower two rows, using preheating obtains good segmentation at the very beginning SA iteration, while not using preheating struggles with it in the first two SA iterations. Thus our query preheating really smooths SA iterations on the image. And it should be the same on the video's first frame.

**Transform differentiation smooths SA recurrences**

The non-informative and informative queries of the first and non-first video frames generally require different transform capabilities through the SA recurrences, so we expect our transform differentiation provides better gradient signals during training. Please refer to Section A.4 for mathematical analysis. To statistically analyze this, we should count the per-frame gradients of the SA module, contributed by per-frame decoding. But in practice, such gradients are always merged together by mainstream deep learning libraries like PyTorch. Thus we take the per-frame gradients of per-frame input features as an indirect reflection. The gradient map is calculated by averaging the per-frame gradients' absolute along the channel dimension, leaving the spatial dimensions for visualization.

As shown in Figure 4 (*right*), we visualize the gradient maps with and without our transform differentiation given a video sample. The input features of both first and non-first frames receive more balanced gradient signals if using transform differentiation than not. Specifically, the gradient maps show better object contours and overall amplitudes with transform differentiation, while showing very unclear object contours and fluctuated amplitudes without it.

## 6 CONCLUSION

In this work, we propose a novel method SmoothSA, which addresses the query cold-start issue in SA iterations on the image or video's first frame, and transform homogeneity issue in SA recurrences across video's first and non-first frames. We introduce two techniques, query preheating and transform differentiating, to address these two issues. With our SmoothSA, OCL models on image and videos achieve new state-of-the-art performance on object discovery, which also benefits downstream tasks including object recognition and visual question answering.

**Limitations and future works**. Although not observed yet, intuitively our method has two possible limitations. For query preheating, if the aggregated slots are bad then the preheated queries are bad and in turn the slots can be even worse; For transform differentiating, we empirically use three and one iterations, instead of automatically, which may not fit all cases. Determining when these two limitations are prominent and how to overcome them are left for future works.

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

# A    APPENDIX

## A.1    LLM USAGE STATEMENT

We used GPT-based tools solely for correcting grammar and improving the readability of the manuscript. No part of the research ideation, experimental design, analysis, or substantive writing was generated by LLMs.

## A.2    ABLATION STUDY

We conduct ablation studies as shown in Table 5.

(*a*) **Query preheating related**:

(*a.1*) Implementing our preheater as a Transformer decoder block is better than as a Slot Attention module;

(*a.1.1*) If using a Transformer decoder block as preheater, then switch the self-attention and cross-attention in it is better than not;

(*a.2*) Stopping gradient on preheated queries is better than not;

(*a.3*) Setting preheating loss weight to 100 is better than other values;

(*b*) **Transform differentiating related**:

(*b.1*) Using shared module weights on first-frame transform $\mathbf{\Phi}_a$ and non-first-frame transforms $\mathbf{\Phi}'_a$ is better than using separate weights;

(*b.2*) For conditioned video OCL, using iteration numbers of 3 and 1 on first and non-first frames respectively is better than other combinations;

(*b.3*) For unconditioned video OCL, using iteration numbers of 3 and 1 on first and non-first frames respectively is better than other combinations.

| | ARI + ARI$_{\text{fg}}$ |
|---|---|
| **Preheater implementation @COCO** | |
| a Transformer decoder block | 68.3$_{\pm0.8}$ |
| a Slot Attention module | 63.3$_{\pm1.4}$ |
| MLP as the preheater | 47.8$_{\pm8.5}$ |
| no preheater and preheat loss | 56.9$_{\pm2.5}$ |
| **Switch cross-attention and self-attention in preheater @COCO** | |
| Yes | 68.3$_{\pm0.8}$ |
| No | 49.6$_{\pm9.4}$ |
| **Stop gradient on preheated query @COCO** | |
| Yes | 68.3$_{\pm0.8}$ |
| No | 67.5$_{\pm2.9}$ |
| **Preheating loss weight @COCO** | |
| 10 | 59.7$_{\pm1.0}$ |
| 50 | 65.5$_{\pm0.4}$ |
| 100 | 68.3$_{\pm0.8}$ |
| 200 | 67.4$_{\pm1.3}$ |
| **Use separate weights for first and non-first transforms @YTVIS** | |
| separate | 52.3$_{\pm0.7}$ |
| shared | 68.3$_{\pm0.8}$ |
| **Unconditional video OCL: first and non-first SA #iter @YTVIS** | |
| 3+1 | 105.6$_{\pm2.2}$ |
| 1+1 | 97.4$_{\pm11.4}$ |
| 3+3 | 103.4$_{\pm6.8}$ |
| **Conditional video OCL: first and non-first SA #iter @MOVi-C** | |
| 3+1 | 136.3$_{\pm7.1}$ |
| 1+1 | 133.9$_{\pm15.0}$ |
| 3+3 | 132.7$_{\pm8.4}$ |

Table 5: Ablation studies.

## A.3  WHY SWAPPING SELF-ATTENTION WITH CROSS-ATTENTION?

Denote self-attention as [sa] while cross-attention as [ca]. The standard Transformer decoder block has the architecture of [sa]-[ca]-[mlp], while our swapped Transformer decoder block has the architecture of [ca]-[sa]-[mlp]. Note that short-cut connections are ignored for simplicity. We use PCA (Principle Component Analysis) to visualize the intermediate slots inside (*i*) the standard Transformer decoder block, i.e., queries and preheated queries after the [sa] as the first module; and (*ii*) the swapped Transformer decoder block, i.e., queries and the preheated queries after [ca] as the first module. The model checkpoints are reused from Table 5.

As shown, our swapped attention produces diverse points, i.e., expressive representations.

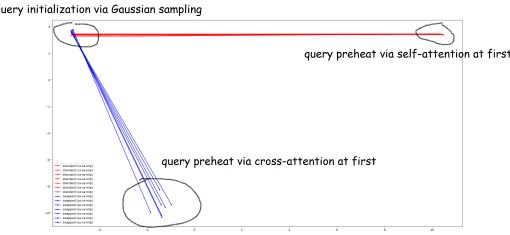

- the queries after initialization – *query initialization after Gaussian sampling*
  - (top-left): clustered together

- and the queries after the first attention
  - for standard transformer decoder block as the preheater, the first attention module is self attention – *query preheating after self-attention at first*
    * (top-right): still clustered together
  - for swapped transformer decoder block as the preheater, the first attention module is cross attention – *query preheating after cross-attention at first*
    * (bottom): become well separated

We explain the observed performance gap and visualization as below:

- The queries before preheating are sampled randomly from some learnt Gaussian distributions (the mainstream case), containing no specific information about current specific features.
- Thus using self-attention to mix them is meaningless, as this still does not introduce any specific information about current specific visual feature.
- Thus inside our preheater, we should first inject the current specific information into the queries by cross attention, then further transformation like self-attention and MLP could be meaningful.

## A.4 MATHEMATICAL ANALYSIS

**Benefit of Preheating**

Follow the settings and notations from Sections 3.1 and 3.2. Although $\phi_a$ and $\phi_p$ take $\boldsymbol{F}$ as the second input argument, we ignore it for simplicity.

In practice, for any fixed $\boldsymbol{F}$, $\phi_a$ is usually a contraction, thus for all $\boldsymbol{X}, \boldsymbol{Y} \in \mathbb{R}^{n \times c}$,

$$||\phi_a^i(\boldsymbol{X}) - \phi_a^i(\boldsymbol{Y})|| \leq \alpha^i ||\boldsymbol{X} - \boldsymbol{Y}|| \quad \text{where } \alpha \in [0, 1) \tag{9}$$

By the Banach fixed point theorem, there is a unique fixed point $\boldsymbol{S}^* = \phi_a(\boldsymbol{S}^*)$, and for every $\boldsymbol{X}$,

$$||\phi_a^i(\boldsymbol{X}) - \boldsymbol{S}^*|| \leq \alpha^i ||\boldsymbol{X} - \boldsymbol{S}^*|| \tag{10}$$

With our preheater, $\phi_p(\boldsymbol{Q})$ is closer to $\boldsymbol{S}^*$ than $\boldsymbol{Q}$, then:

$$||\phi_p(\boldsymbol{Q}) - \boldsymbol{S}^*|| \leq q ||\boldsymbol{Q} - \boldsymbol{S}^*|| \quad \text{where } q \in [0, 1) \tag{11}$$

And then:

$$||\phi_a^3(\phi_p(\boldsymbol{Q})) - \boldsymbol{S}^*|| \leq \alpha^3 ||\phi_p(\boldsymbol{Q}) - \boldsymbol{S}^*|| \leq q\alpha^3 ||\boldsymbol{Q} - \boldsymbol{S}^*|| \tag{12}$$

Compared with not using preheater,

$$||\phi_a^3(\boldsymbol{Q}) - \boldsymbol{S}^*|| \leq \alpha^3 ||\boldsymbol{Q} - \boldsymbol{S}^*|| \tag{13}$$

Therefore, the preheated run is strictly closer to the fixed point after three iterations than the non-preheated run would be.

**Benefit of Differentiating**

Follow the settings and notations from Sections 3.1 and 3.3. Although $\phi_a$ takes $\boldsymbol{F}$ as the second input argument, we ignore it for simplicity. We treat one frame at a time, dropping subscript $t$.

We also supplement the decoding part of OCL here, where reconstruction is utilized for supervision. Reconstruction is $\boldsymbol{X}' = \phi_d(\boldsymbol{S})$ and loss is $l = \text{MSE}(\boldsymbol{X}', \boldsymbol{X})$.

We follow the assumption of Equation (9).

According to Lipschitz Jacobian bounds,

$$||\frac{\partial \phi_a(\boldsymbol{S})}{\partial \boldsymbol{S}}|| \leq \alpha \quad \text{(consistent with contraction)} \tag{14}$$

$$||\frac{\partial \phi_a(\boldsymbol{S})}{\partial \theta_a}|| \le B \quad \text{for all } \boldsymbol{S} \text{ (bound on how strong parameters in each iteration)} \tag{15}$$

$$||\frac{\partial \phi_d(\boldsymbol{S})}{\partial \boldsymbol{S}}|| \le L \quad \text{(bound on the largest loss)} \tag{16}$$

Unroll the total $i_1$ iterations. Let $\boldsymbol{J}_i := \frac{\partial \phi_a(\boldsymbol{S}^{(i-1)})}{\partial \boldsymbol{S}}$ and $\boldsymbol{U}_i := \frac{\partial \phi_a(\boldsymbol{S}^{(i-1)})}{\partial \theta_a}$, $\boldsymbol{D} := \frac{\partial \phi_d(\boldsymbol{S})}{\partial \boldsymbol{S}}$ and $\boldsymbol{G_X} := \frac{\partial l}{\partial \boldsymbol{X}'}$. And the derivative of the final $\boldsymbol{S}^{(i_1)}$ w.r.t $\theta_a$ is the sum of contributions from each unrolled iteration:

$$\frac{\boldsymbol{S}^{(i_1)}}{\partial \theta_a} = \Sigma_{i=1}^{i_1}(\Pi_{m=i+1}^{i_1}\boldsymbol{J}_m)\boldsymbol{U}_i \tag{17}$$

By chain rule, the full gradient is

$$\frac{\partial l}{\partial \theta_a} = \boldsymbol{G_X}^T\boldsymbol{D}\frac{\partial \boldsymbol{S}^{(i_1)}}{\partial \theta_a} = \boldsymbol{G_X}^T\boldsymbol{D}\Sigma_{i=1}^{i_1}(\Pi_{m=i+1}^{i_1}\boldsymbol{J}_m)\boldsymbol{U}_i \tag{18}$$

For the frame with $i_1$ iterations,

$$||\frac{\partial l}{\partial \phi_a}|| \le ||\boldsymbol{G_X}||LB\Sigma_{i=0}^{i_1-1}\alpha^i = ||\boldsymbol{G_X}||LB\frac{1-\alpha^{i_1}}{1-\alpha} < ||\boldsymbol{G_X}||LB\frac{1}{1-\alpha} \tag{19}$$

where on the right side, only $||\boldsymbol{G_X}||$ depends on the number of iterations $i_1$.

In practice, for the first frame $||\boldsymbol{G_X}||$ tends to be large and more iterations reduces it, while for non-first frames $||\boldsymbol{G_X}||$ tends to be small and less iterations are needed. And if we insist to use more iterations for non-first frames (the same number of iterations as the first frame), $||\boldsymbol{G_X}||$ can be too small compared with that of the first frame. This causes imbalanced gradient contributions from the first and non-first frames to $\phi_a$ parameters.

## A.5 VIDEO OCL ON YTVIS21 USING ORIGINAL VIDEO LENGTH

Here are the object discovery results on dataset YTVIS21[12]. Note that the results are produced on SlotContrast official codebase[13], namely, we adopt all the hyperparameters used by SlotContrast, especially using the original video length, instead of clipping them to constant length 20 as in Table 2. Also note that the metrics $ARI_{fg}$ and mBO are the video ones, rather than the image ones.

| @YTVIS21 | video ARIfg | video mBO |
|---|---|---|
| VideoSAUR (copied values) | 28.9 | 26.3 |
| SlotContrast (copied values) | 38.0 | 33.7 |
| SmoothSA (seed@42,43,44) | 45.9±1.2 | 36.7±0.5 |

## A.6 COMPUTATION OVERHEAD

We provide concrete numbers of the computation overhead below. Our method shows better computation efficiency in both training time and memory consumption than baselines.

| V100 | training time / hours | memory consumption / GB |
|---|---|---|
| SPOT @COCO, bs32 | 4.7 | 8.5 |
| DIAS @COCO, bs32 | 4.5 | 9.4 |
| SmoothSA @COCO, bs32 | 4.2 | 8.7 |
| SlotContrast @YTVIS, bs8 | 7.4 | 6.4 |
| RandSF.Q @YTVIS, bs8 | 6.8 | 5.2 |
| SmoothSA @YTVIS, bs8 | 7.1 | 5.2 |

---

[12]https://youtube-vos.org/challenge/2021

[13]https://github.com/martius-lab/slotcontrast

## A.7 RESULTS ON DATASET OVIS

We also evaluate our method's effectiveness on more challenging datasets. We choose OVIS[14] (Qi et al., 2022), which is more occluded than YTVIS. Our method still shows some superiority over the baseline. But the performance boosts over the baseline is much smaller, compared with the results of dataset YTVIS in Table 2.

| @OVIS | ARI | ARIfg | mBO | mIoU |
|---|---|---|---|---|
| SlotContrast | 37.2±1.0 | 44.8±1.0 | 20.1±0.8 | 17.1±0.9 |
| SmoothSA | 39.3±2.3 | 47.5±0.4 | 21.0±0.4 | 19.5±0.4 |

---

[14]https://songbai.site/ovis

