# OpenReview forum: "Smoothing Slot Attention Iterations and Recurrences"
_ICLR.cc/2026/Conference — ICLR 2026 Conference Withdrawn Submission_

### Official Review · Reviewer_xdve · 2025-10-30

**Soundness:** 3
**Presentation:** 3
**Contribution:** 2
**Rating:** 4
**Confidence:** 3

**Summary:**

This paper introduces SmoothSA, a novel method to enhance Slot Attention (SA) in object-centric learning for images and videos.  Its main contributions are: 1. addressing the query cold-start issue by preheating queries with informative features, and 2. tackling the transform homogeneity issue across video frames by differentiating transform operations for first and subsequent frames.

**Strengths:**

1. The paper addresses two long-overlooked limitations in mainstream Object-Centric Learning with Slot Attention and introduces two simple yet impactful techniques with minimal computational overhead to solve the problems: 1). Query Preheating; 2). Differentiated Transforms.
2. The method is extensively evaluated across multiple tasks, datasets, and baselines, and achieves consistent performance.

**Weaknesses:**

1. The choice of 3 iterations for the first frame and 1 for non-first frames is empirical, not adaptive. The paper acknowledges this may not generalize to all scenarios, but it does not explore dynamic iteration adjustment.
2. While ablations show a Transformer decoder block (with swapped attention) is effective for preheating, the paper does not compare it to other lightweight architectures (e.g., MLPs) or analyze in detail why swapped attention outperforms standard attention (considering that in Table 5 the performance gap of swapping is huge).
3. Paper should provide a more detailed comparison of the computation overhead of each designed component.
4. The experiments are mainly conducted on standard and rather easy scenarios, and I wonder the model's effectiveness on more challenging scenarios. In VIS for example, author should test on OVIS, which is more occluded than YTVIS.

**Questions:**

See weaknesses.

---

> ### Author Response · Authors · 2025-11-19
>
> Thank you for the valuable feedback!
>
> ---
>
>
> # `W1` Dynamic Number of Iterations
> > The choice of 3 iterations for the first frame and 1 for non-first frames is empirical, not adaptive. The paper acknowledges this may not generalize to all scenarios, but it does not explore dynamic iteration adjustment.
>
> **We agree** that dynamic #iterations is conceptually better than empirical #iterations, but it **might only yield marginal** performance gains according to Table 5 Lines 615-622 -- Different #iterations have limited performance gaps. Note: the numbers are the summation of ARI and ARIfg, which roughly doubles performance gaps.
>
>
> # `W2` Compare with MLP; Why Swapped Attention Better
> > compare with other lightweight architectures, e.g., MLPs; analyze why swapped attention outperforms standard attention
>
> Following your advice, we conduct an **additional ablation of MLP as the preheater**. MLP can not take both features and queries together as input, thus its performance is inferior.
>
> |                @COCO                 | ARI+ARIfg |
> | :----------------------------------: | :-------: |
> | our proposed design as the preheater | 68.3±0.8  |
> |         **MLP as the preheater**         | 47.8±8.5  |
> |   no preheater and preheating loss   | 56.9±2.5  |
>
> Such information is updated into `Lines 598-599` of our revised paper.
>
> Thank you for suggesting us analyzing **why *swapped* attention is much better than *standard***. We use PCA (Principle Component Analysis) to visualize:
> - the queries after initialization -- `query initialization after Gaussian sampling`
>   - (*top-left*): clustered together
> - and the queries after the first attention
>   - for ***standard** transformer decoder block* as the preheater, the first attention module is *self attention* -- `query preheating after self-attention at first`
>     - (*top-right*): still clustered together
>   - for ***swapped** transformer decoder block* as the preheater, the first attention module is *cross attention* -- `query preheating after cross-attention at first`
>     - (*bottom*): become well separated
>
> As shown in the figure below, apparently, our swapped attention gives much diverse points, i.e., expressive representations.
>
> [[figure](https://anonymous.4open.science/r/smoothsa_rebuttal-EC8C/swapped_transformer_decoder_block.png)] https://anonymous.4open.science/r/smoothsa_rebuttal-EC8C/swapped_transformer_decoder_block.png
>
> **We explain this as below**:
> - The queries before preheating are sampled randomly from some learnt Gaussian distributions (the mainstream case), containing no specific information about current specific features.
> - Thus using self-attention to mix them is meaningless, as this still does not introduce any specific information about current specific visual feature.
> - Thus inside our preheater, we should first inject the current specific information into the queries by cross attention, then further transformation like self-attention and MLP could be meaningful.
>
> Such information is updated into `Appendix A.3`, i.e., `Lines 627-666` of our revised paper.
>
>
> # `W3` Computation Overhead Details
> > detailed comparison of the computation overhead
>
> Thank you for your suggestion. We provide concrete numbers below. As you can see, our method shows better computation efficiency in both *training time* and *memory consumption* than baselines.
>
> |                V100 | training time / hours | memory consumption / GB |
> | ------------------: | :-------------------: | :---------------------: |
> |          SPOT @COCO, bs32 |          4.7          |           8.5           |
> |          DIAS @COCO, bs32 |          4.5          |           9.4           |
> |  **SmoothSA$^i$** @COCO, bs32 |          4.2          |           8.7           |
> | SlotContrast @YTVIS, bs8 |          7.4          |           6.4           |
> |     RandSF.Q @YTVIS, bs8 |          6.8          |           5.2           |
> | **SmoothSA$^v$** @YTVIS, bs8 |          7.1          |           5.2           |
>
> Such information is updated into `Appendix A.6`, i.e., `Lines 742-753` of our revised paper.

---

> > ### Author Response · Authors · 2025-11-27
> >
> > # `W4` Results on Dataset OVIS
> > > Should test on OVIS, which is more occluded than YTVIS.
> >
> > Thank you for sharing such a more up-to-date benchmark. **Our method still shows some superiority** over the baseline. But the performance boosts over the baseline is much **smaller**, compared with the results of dataset YTVIS in Table 2.
> >
> > |    @OVIS     |   ARI    |  ARIfg   |   mBO    |   mIoU   |
> > | :----------: | :------: | :------: | :------: | :------: |
> > | SlotContrast | 37.2±1.0 | 44.8±1.0 | 20.1±0.8 | 17.1±0.9 |
> > | **SmoothSA$^v$** | 39.3±2.3 | 47.5±0.4 | 21.0±0.4 | 19.5±0.4 |
> >
> > Please note:
> > - The segmentation annotation for the val/test subsets are not available, thus we train the models on the val/test subsets while evaluate on the train subset;
> > - The official evaluation server uses metric AP (*classification*+*segmentation*), which is not compatible with video OCL tasks, i.e., unsupervised *segmentation*, thus we evaluate with metrics ARI, ARIfg, mBO and mIoU.
> >
> > Such information is updated into `Appendix A.7`, i.e., `Lines 756-765` of our revised paper.
> >
> >
> > ---
> >
> > Hope all your concerns are addressed.
> > We would be grateful if you could re-evaluate our work accordingly.
> > Please let us know if you have any further questions.

---

### Official Review · Reviewer_eLvL · 2025-11-01

**Soundness:** 2
**Presentation:** 3
**Contribution:** 2
**Rating:** 2
**Confidence:** 4

**Summary:**

The paper presents an essential method to smoothen slot attention iterations SmoothSA which aims to improvise on existing slot attention and provides a solutions to smoothen the predictions. Core contributions include addressing cold start issue as in noted in the paper.

**Strengths:**

The papers to claim that they are the first in addressing the query cold start issue in SA. While the problem sounds interesting I still have reservations.

The authors claim to solve for the first time addressing the transform homogeneity issue in SA recurrences across the first and non-first frames

Results seems competitive in comparison to previous methods

**Weaknesses:**

1. c1 needs more basis on why this issue needs to be looked at. While the authors say best of their knowledge. I am not convinced by this, is there any basis based on experiments, notes from other papers, any other proof documented?

2. I believe working on query initializations has been active as in indicated in BO-QSA, meta slot, https://arxiv.org/abs/2404.19654 also uses multiple queries for inference which boosts performance. It will be therefore interesting and a requirement on how the current strategy acts against the current method

3. c3 and c4 are virtually the same I do not see why they should be separate points

4. The experiments lack comparison to SOTA methods in SA, every table is almost compared to 1 method which I feel is not enough

5. The discussions lack references, it seems like this paper pretty much solves a lot of things even though no one else noted anything in the past

6. Comment 2 say as heavy RNN modules, in practice slot attention uses GRU which is not that heavy in theory, I do not get this point. Also Slot attention is independent of any overall encoding and decoding. More experiments and concrete numbers with results are needed to show this point

7. The analysis lacks any ablations and concrete discussions. The current discussions are too generic to show any point. This gives me a sense of unfinished work and early submission than intended

**Questions:**

See weaknesses

**Details Of Ethics Concerns:**

I believe LLMS have been used to draw conclusions analysis and inferences which is more than what the llm usage statement says

---

> ### Author Response · Authors · 2025-11-19
>
> Thank you for your feedback.
>
>
> ---
>
> # `W1` Contribution *c1* Not Convincing
> > c1 needs more basis on why this issue needs to be looked at. While the authors say best of their knowledge. I am not convinced by this, is there any basis based on experiments, notes from other papers, any other proof documented?
>
> Please **check Lines 055~062** in our paper: The facts behind the two issues we address, namely *c1* and *c2*, have been indirectly presented in prior works like [BO-QSA] and [STEVE].
>
> Please **check Lines 079~081** in our paper: What we claim is the first to address these issues, rather than the first to present the facts behind these issues.
>
>
> # `W2` Our Method vs Query Initialization
> > Query initializations has been active as in indicated in BO-QSA and MetaSlot.
> > It is interesting how the current strategy acts against these methods.
>
> As we have discussed in **Lines 106~111** in our paper, our method operates between query initialization and Slot Attention aggregation, thus is *orthogonal* to BO-QSA and MetaSlot. As shown below, our SmoothSA can improve BO-QSA and MetaSlot.
>
> |                 @COCO |   ARI    |  ARIfg   |   mBO    | mIoU     |
> | --------------------: | :------: | :------: | :------: | -------- |
> |                BO-QSA | 21.2±0.8 | 39.0±0.9 | 29.5±1.0 | 28.4±0.6 |
> |   **SmoothSA$^i$**+BO-QSA | 25.7±2.1 | 41.5±0.9 | 32.8±0.6 | 31.4±0.5 |
> |              MetaSlot | 23.4±1.8 | 40.4±0.6 | 29.7±0.4 | 27.9±0.3 |
> | **SmoothSA$^i$**+MetaSlot | 26.7±3.2 | 42.6±1.4 | 33.4±1.5 | 31.5±0.7 |
>
>
> # `W3` Why Not Merging Contributions *c3* and *c4*
> > *c3* and *c4* are virtually the same
>
> We separated them to **emphasize our contribution to the downstream advanced vision tasks**, like *Object Recognition*, *Visual Reasoning* or *Visual Question Answering* on images and videos, as most OCL papers only report results on OCL benchmarks. This can be verified by skimming some OCL papers, e.g., BO-QSA and MetaSlot you mentioned.
>
>
> # `W4` Lacking SotA; Only 1 Baseline
> > The experiments lack comparison to SOTA methods in SA, every table is almost compared to 1 method which I feel is not enough.
>
> Please **check the tables** in our paper, especially Table 1 and Table 2, where **4 and 5 baselines are included**, even the **most recent SotA methods** published in ACM MM 2025 (DIAS) and AAAI 2026 (RandSF.Q).
>
>
> # `W5` Solve Many Things No One Noted Before
> > The discussions lack references, it seems like this paper pretty much solves a lot of things even though no one else noted anything in the past.
>
> Please **check Lines 424 and 441** in our paper: The two headlines in Section 5 Discussion. We addressed just *two issues*, not "a lot of things".
>
> Please **check Lines 079~081** in our paper: What we claim is the first to address these issues, rather than the first to notice the facts behind these issues.
>
>
> # `W6` Concrete Numbers for Computation Overhead
> > More experiments and concrete numbers with results are needed to show (computation overhead).
>
> Please refer to our response to Reviewer `xdve`'s review `W3`.
>
>
> # `W7` Lacks Ablation and Concrete Discussion => Unfinished Work
> > The analysis lacks any ablations and concrete discussions. The current discussions are too generic to show any point. This gives me a sense of unfinished work and early submission than intended.
>
> For ablation studies, please **refer to Table 5**. For concrete discussions, please refer to **Section 5 Discussion** visual analyses, as well as **Appendix A.3** mathematical analyses. We believe our visual plus mathematical analyses are already quite concrete.
>
> # `W8` Overusing LLM
> > I believe LLMs have been used to draw conclusions analysis and inferences, which is more than what the LLM Usage Statement says.
>
> As stated in the paper, we only used LLM for correcting grammar and improving the readability. **You can verify this by checking our text using any AI text detectors**.
>
>
> ---
>
> Hope all your concerns are addressed.
> We would be grateful if you could re-evaluate our work accordingly.
> Please let us know if you have any further questions.

---

> > ### Comment · Reviewer_eLvL · 2025-11-20
> >
> > Thanks for the response
> >
> > w1: Could the authors point to any specific lines in their paper where they show that what exactly are the issues? It seems like the lines ```Such queries contain no information about any specific sample, thus to obtain slots by refining queries using SA on image features, typically three iterations are necessary.``` Do the authors have any analysis that correlates to this? Because it is not very apparent that the problems faced by Jia et al. 2023 is really due to the cold start issue? The rationale behind the intuition is missing it appears to me
> >
> > w2: Before moving on, are there only two methods that has till date worked on Query initializations on SA?
> >
> > w3: I believe that they are the same or in a way connected to each other and is redundant and is a bit assertive to establish this paper. This seems a bit overstating of the experimental contributions
> >
> > w4: Tab 3 and Tab 4 seems to have been compared to only one baseline. Any justifications?
> >
> > w5: Though I am now convinced the authors focus on two issues but the background to establish those seems lacking. Fundamental questions like why they are required at the first place, how directly or indirectly other methods have attempted to solve previously seems missing in the first place
> >
> > w6: Please also specify the hardware and software overheads
> >
> > w7: See comments above

---

> ### Author Response · Authors · 2025-11-25
> **Consensus With Reviewer `eLvL` After Clarification**
>
> Before addressing your new concerns, we briefly restate key points from our earlier discussion. **Since no further objections were raised, we understand these concerns to be resolved** and list them here for shared clarity moving forward.
>
>
> # `W1.a`
>
> We are the first to address the *query cold-start* issue, whereas prior works like BO-QSA and MetaSlot only indirectly present the *facts* behind this issue.
>
>
> # `W2.a.1`
>
> Our method works well with different *query initialization* techniques, like BO-QSA and MetaSlot.
>
>
> # `W2.a.2`
>
> BO-QSA and MetaSlot target the *query initialization* issue, while we address the *query cold-start*/*preheating* issue.
>
>
> # `W3.a`
>
> Most OCL works do not cover *downstream advanced vision tasks* like VQA.
>
>
> # `W4.a`
>
> Within the scope of OCL itself, our experiments do cover both *multiple baselines* and the *latest SotA* methods.
>
>
> # `W5.a`
>
> We address *two issues*, not "*a lot of things*". The necessities and benefits are analyzed both *visually* (`Section 5`) and *mathematically* (`Appendix A.3`).
>
>
> # `W6.a`
>
> We have provided the *computation overhead*, i.e., training time and memory consumption.
>
>
> # `W7.a`
>
> We did provide *ablation* studies and *visual* / *mathematical analyses* in the original submission. And our submission is not an *unfinished work*.
>
>
> # `W8.a`
>
> Our paper contains no ethics issues like *LLM overuse*.

---

> ### Author Response · Authors · 2025-11-25
> **Further Clarification Requested by Reviewer `eLvL`**
>
> # `W1.b`
> > Could the authors point to any specific lines in their paper?
>
> Please refer to *Section 3.1* the *first paragraph* of paper BO-QSA, i.e., "*such random initializations provide no hint on the notion of object*". This is what we *define* as *query cold-start*. This also aligns with our statements in the paper, e.g., in `Lines 064` - `065`, "*the cold-start queries lack scene-specific information*".
>
>
> # `W2.b`
> > Are there only two methods that work on Query initialization?
>
> Yes. We searched Google Scholar by keywords `"object-centric learning" "query initialization"` and found ONLY the following papers:
>
> - `BO-QSA` Improving Object-centric Learning with Query Optimization
>   - address Q init directly
> - Temporally consistent object-centric learning by contrasting slots
>   - not address Q init
> - `MetaSlot` Break Through the Fixed Number of Slots in Object-Centric Learning
>   - address Q init directly
> - Bootstrapping top-down information for self-modulating slot attention
>   - not address Q init
> - Multi-Scale Fusion for Object Representation
>   - not address Q init
> - Dynamic Category Queries Transformer for Generalized Few-shot Semantic Segmentation
>   - not related
> - Online Long-term Point Tracking in the Foundation Model Era
>   - not related
> - Conditional set generation with transformers
>   - not related
> - Track-On2: Enhancing Online Point Tracking with Memory
>   - not related
> - Style transformer for image inversion and editing
>   - not related
>
>
> # `W3.b`
> > Contributions c3 and c4 are the same or in a way connected to each other.
>
> We agree, and we merge c3 and c4 into one contribution:
>
> *(Our contributions are ...) (c1) ... (c2) ... (c3) new state-of-the-art on both image and video OCL benchmarks, along with consistent performance gains on downstream advanced vision tasks*.
>
>
> # `W4.b`
> > Tab 3 and Tab 4 seem to have been compared to only one baseline.
>
> This is true. However, as you and us agreed in `W3.a` and `W4.a`, *most OCL works do not cover any downstream tasks* like ours. Adding more downstream baselines contributes little to our theme of Object-Centric Learning.
>
> Still, we provide more results:
>
> (1) More results on **Object Recognition**
> |                    | class top1 |      bbox R2      |
> |:------------------:|:----------:|:-----------------:|
> | **image obj recognition** |    COCO    |      #slot=7      |
> |     SPOT + MLP     |  0.67±0.0  |      0.62±0.1     |
> |     DIAS + MLP     |  0.72±0.1  |      0.62±0.1     |
> | SmoothSA$^i$ + MLP |  0.73±0.0  |      0.64±0.1     |
> | **video obj recognition** |    YTVIS   | #slot=7, #step=20 |
> | SlotContrast + MLP |  0.40±0.1  |      0.53±0.1     |
> |   RandSF.Q + MLP   |  0.49±0.0  |      0.60±0.1     |
> | SmoothSA$^v$ + MLP |  0.50±0.0  |      0.62±0.0     |
>
> (2) More results on **Visual Question Answering**
> |      image VQA      |      GQA     |     #slot=7    |
> |:-------------------:|:------------:|:--------------:|
> |                     |  accuracy %  |                |
> |     SPOT + Aloe     |   52.3±2.8   |                |
> |     DIAS + Aloe     |   55.7±1.4   |                |
> | SmoothSA$^i$ + Aloe |   56.7±1.9   |                |
> |    **video VQA**    |  **CLEVRER** |   **#slot=7**  |
> |                     | per option % | per question % |
> | SlotContrast + Aloe |   97.2±1.1   |    95.6±0.9    |
> |   RandSF.Q + Aloe   |   98.0±0.8   |    96.3±1.0    |
> | SmoothSA$^v$ + Aloe |   98.7±0.4   |    96.9±0.6    |
>
> # `W5.b`
> > Why they are required? How other methods have attempted to solve previously?
>
> As you and us agreed in `W5.a`, without addressing these two issues, image OCL suffers from limited object discovery through typical SA iterations (`Figure 4` left, `Equation 13`), while video OCL suffers from unbalanced gradient propagation across typical SA recurrences (`Figure 4` right, `Equation 19`).
>
> As you and us agreed in `W1.a` and `W2.a.2`, previous methods address only *query initialization*, not *query preheating* as we do.
>
>
> # `W6.b`
> > Also specify the hardware and software overheads.
>
> As you and us agreed in `W6.a`, we have already added training time and memory consumption as the "*hardware overhead*".
>
> As for "***software overhead***", we could not find this term in the OCL literature. Could you please clarify what specific kind of overhead you are referring to?
>
>
> # `W7.b`
> > See comments above
>
> As you and us agreed in `W7.a`, we have provided *visual* and *mathematical* analyses. Could you please clarify what additional analyses you would like to see?

---

### Official Review · Reviewer_FgBa · 2025-11-08

**Soundness:** 3
**Presentation:** 3
**Contribution:** 3
**Rating:** 4
**Confidence:** 4

**Summary:**

The paper targets two frictions in Slot Attention (SA)–based object-centric learning (OCL): (1) cold-start queries on the first image/video frame, and (2) homogeneous aggregation transforms applied identically across all video frames despite their differing query conditions. The authors propose SmoothSA with two simple modifications: a tiny preheater that “preheats” first-frame queries using input features via self-distillation; and a differentiated recurrence policy using full three SA iterations on the first frame and a single iteration on non-first frames. Experiments on image (CLEVRtex, COCO, VOC) and video (YTVIS, VideoSAUR) OCL, plus downstream object recognition and VQA, show consistent gains over strong baselines such as SPOT and SlotContrast.

**Strengths:**

Clear identification of two practical bottlenecks in SA pipelines (first-frame cold start; transform homogeneity across frames), with a minimal, easy-to-adopt solution.

Method simplicity: the preheater is a light Transformer decoder–style module trained via self-distillation inside the OCL model; the recurrence rule (3 iterations on the first frame, 1 on others) is plug-and-play.

Broad empirical coverage: improvements reported on synthetic and real-world image datasets (ARI/ARI_fg/mIoU/mBO) and video datasets, with qualitative masks and downstream boosts in object recognition and VQA (GQA, CLEVRER).

Positioning vs prior work is knowledgeable (e.g., BO-QSA, MetaSlot, SAVi/++, SlotContrast, STATM, SlotPi, RandSF.Q). The paper argues SmoothSA addresses issues orthogonal to prior query-initialization or query-prediction lines.

**Weaknesses:**

1. Unclear baseline performances:
The reported results for VideoSAUR and SlotContrast are substantially higher than those in the original papers, particularly in terms of FG-ARI. This discrepancy raises questions about the experimental comparability—whether the authors re-implemented these methods under different settings, used stronger backbones, or applied additional training tricks. A clear explanation is needed to ensure that the reported improvements are attributable to the proposed method rather than to differences in baseline implementations.

2. Inconsistent evaluation metrics:
The paper evaluates SmoothSA on both image and video datasets but seems to use  same metrics for them. In image datasets, the image FG-ARI measures spatial clustering quality, while in video datasets the video FG-ARI captures both spatial segmentation and temporal consistency. However, the paper interprets these results uniformly, without clarifying the differences. This makes it difficult to fairly assess the improvements and understand whether the gains arise from better object segmentation, temporal stability, or both.

3. Unclear experimental setup and sequence length generalization.
The paper does not specify the segment lengths used during training and testing. In video object-centric learning, training typically uses short clips while testing involves longer sequences—making sequence length generalization a central challenge. The absence of this information obscures how SmoothSA performs under distribution shifts in sequence length, and whether its recurrence design truly enhances temporal robustness.

**Questions:**

The main questions are already reflected in the Weaknesses section above, concerning (1) the discrepancy between reported and original baseline performances, (2) the unclear interpretation of evaluation metrics across image and video datasets, and (3) the lack of clarity regarding training/testing segment lengths and sequence length generalization.

---

> ### Author Response · Authors · 2025-11-19
>
> Thank you for the valuable feedback!
>
>
> ---
>
> # `W1` Substantially Higher Baseline Performance
> > The reported results for VideoSAUR and SlotContrast are substantially higher than those in the original papers, particularly in terms of FG-ARI.
>
> Short answer: We **clip all videos into a shorter constant length**, i.e., `20`, instead of using their original variant lengths, i.e., `20~70`. As is known, shorter video leads to higher OCL performance.
>
> Crucially, **all methods are experimented under identical configs**, including *advanced data augmentation*, *backbones* and *other training tricks*. This can be verified by comparing any two config files in our supplemental file.
> For example:
> - `log/smoothsa_r-coco/smoothsa_r-coco.py` vs `log/spot_r-coco/spot_r-coco.py`
>   - Mind the `### datum` and `### learn` code sections.
> - `log/smoothsav_r-ytvis/smoothsav_r-ytvis.py` vs `log/slotcontrast_r-ytvis/slotcontrast_r-ytvis.py`
>
> Even **upon the SlotContrast official codebase**, namely, following the SlotContrast original configs, e.g., *using the original video length*, our method still wins out.
>
> |             @YTVIS21             | video ARIfg | video mBO |
> | :------------------------------: | :---------: | :-------: |
> |    VideoSAUR (copied values)     |    28.9     |   26.3    |
> |   SlotContrast (copied values)   |    38.0     |   33.7    |
> | **SmoothSA$^v$** (seed@42,43,44) |  45.9±1.2   | 36.7±0.5  |
>
> Such information is updated into `Appendix A.5`, i.e., `Lines 730-740` of our revised paper.
>
>
> # `W2` Unclear Metrics across Images and Videos
> > In image datasets, the image FG-ARI measures spatial clustering quality, while in video datasets the video FG-ARI captures both spatial segmentation and temporal consistency. However, the paper interprets these results uniformly, without clarifying the differences.
>
> Thank you for reminding us. We definitely agree. We did use *image ARI*/*ARIfg*/*mBO*/*mIoU* for image datasets while *video ARI*/*ARIfg*/*mBO*/*mIoU* for video datasets.
>
> Such information is updated into `Lines 343-346` of our revised paper.
>
>
> # `W3` Unclear Experiment Setup for Sequence Length
> > The paper does not specify the segment lengths used during training and testing.
> > The absence of this information obscures how SmoothSA performs under distribution shifts in sequence length, and whether its recurrence design truly enhances temporal robustness.
>
> The exact training/validation lengths are `5 for training` and `20 for evaluation`, which were adopted in our original submission by all baselines, i.e., VideoSAUR, SlotContrast, RandSF.Q and SmoothSA$^v$ on dataset YTVIS.
>
> These settings are also visible in our supplemental files, e.g.,
> ```shell
> log/
>   slotcontrast_r-ytvis/
>     slotcontrast_r-ytvis.py
>     # dict(type="StridedRandomSlice1", .., size=5),  # length=5 for training
>     # ...
>     # dataset_v = dict(
>     #   type="YTVIS",
>     #   ts=20,  # total length=20; no `slicing`, thus length=20 for validation
>     # ...
>   .../
>     ...py
> ```
>
> Such information is updated into `Lines 343-346` of our revised paper.
>
>
> ---
>
> Hope all your concerns are addressed.
> We would be grateful if you could re-evaluate our work accordingly.
> Please let us know if you have any further questions.

---

### Author Response · Authors · 2025-11-29
**Global Summary for the Area Chair**

Dear Area Chair,

We provide a summary of the rebuttal discussion, given the current exceptional situation. Our goal is to present the state of the dialogue and how all concerns have been addressed.

---

# Reviewer `FgBa` (Rating: 4)

All three concerns have been fully and directly resolved.

## 1. Baseline discrepancy (See `FgBa` `W1`)
- We clarified that higher baseline numbers stem from using a shorter clip length -- a config applied consistently to all methods.
- We also re-ran comparisons following the VideoSAUR/SlotContrast settings preferred by the reviewer. Our SmoothSA still shows significant superiority.
- This has been updated into `Appendix A.5`.

## 2. Metrics clarity (See `FgBa` `W2`)
- We did use the widely-recognized metrics, i.e., the ones suggested by the reviewer.
- We made this explicit in `Lines 343`-`346`.

## 3. Sequence length for train/val (See `FgBa` `W3`)
- We clarified the sequence lengths (5 for training, 20 for validation) and confirmed that all baselines follow identical configs.
- This was added to `Lines 343`-`346`.

No further objections were raised after our clarifications.

---

# Reviewer `xdve` (Rating: 4)

We addressed all four concerns, and conducted additional experiments requested by the reviewer.

## 1. Dynamic #iterations (See `xdve` `W1`)
- We agreed that the suggested dynamic #iterations are conceptually better than our empirical #iterations;
- Our ablations (`Table 5`) show limited performance differences from changing the number of iterations. We therefore retain our simpler empirically validated settings.

## 2. Comparison with MLP; why swapped attentions (See `xdve` `W2`)
- We added ablations showing MLP performs substantially worse.
- We included a PCA analysis (`Appendix A.3`) explaining why swapped attentions produce more expressive queries.
- This was updated into `Lines 598`-`599` and `Lines 627`-`666`.

## 3. Computation overhead (See `xdve` `W3`)
- We added concrete training-time and memory consumption numbers, showing our SmoothSA is actually comparable or more efficient than strong baselines.
- This was updated into `Appendix A.6`.

## 4. More challenging datasets (See `xdve` `W4`)
- Per request, we ran additional experiments on the OVIS dataset, where our SmoothSA also improves over the SlotContrast baseline.
- This was updated into `Appendix A.7`.

All items above were addressed explicitly; no further objections were raised.

---

# Reviewer `eLvL` (Rating: 2)

This reviewer raised multiple concerns based on misunderstandings of the paper. Throughout the discussion, we clarified these misunderstandings with explicit citations to our paper and to related work.

## 1. Misunderstanding examples:
- The reviewer asserted that our tables include only one baseline and exclude state-of-the-art methods. In fact `Tables 1` and `2` include four to five baselines each, including the most recent state-of-the-art. (See `eLvL` `W4`)
- The reviewer claimed we assert "solving many things no one noted before", whereas the paper clearly highlight in `Lines 79`-`81` and `Lines 424` and `441` that we address just two. (See `eLvL` `W5`)
- Although presented in `Table 5` (various ablations) and `Section 5` + `Appendix A.4` (visualization + mathematical analyses), ablations and concrete discussions were claimed to be missing. (See `eLvL` `W7`)
- The reviewer raised ethic issues of LLM overuse, which can be falsified by feeding our paper to any AI text detectors. (See `eLvL` `W8`)

## 2. In our responses, we provided: (See `eLvL` `W1`, `W4`, `W5`, `W7`, `W1.b`, `W2.b` and `W5.b`)
- point-by-point textual references to the exact lines resolving each misunderstanding;
- a detailed list of resolved concerns to establish common ground;
- additional context and experiment results when needed.

## 3. Requests too unspecific to meaningfully address:
- "***software overhead***" without a definition in the literature (see `eLvL` `W6.b`)
- "concrete discussions" (more than our visual and mathematical analyses) without specifying the desired scope (See `eLvL` `W7` and `W7.b`)

Given the above, we believe Reviewer `eLvL`'s score does not reflect the technical content and evidence in the rebuttal.

---

# Overall assessment

- Reviewers `FgBa` and `xdve` raised constructive points that we have fully addressed with clarifications, ablations, and additional experiments.
- Reviewer `eLvL`'s concerns largely reflect factual misunderstandings that we corrected with explicit references and supporting material.

Under normal rebuttal conditions, the discussion would have continued. In the current setting, we respectfully ask the Area Chair to consider this context when making the final decision.

---

Thank you for your time and careful consideration.

The authors

---

### Note · Authors · 2026-01-26

I have read and agree with the venue's withdrawal policy on behalf of myself and my co-authors.

---

### Meta-Review · Area_Chair_nZk9 · 2026-01-05

**Summary:**

The paper proposes two contributions to improve Slot Attention based models for object-centric learning. 1) Addressing the cold-start problem for queries by "preheating" them with some informative features. 2) Addressing the transform homogeneity problem across video frames, by differentiating transformation operations for the first and subsequent frames.

Reviewers pointed out a number of varied concerns about the paper. These include issues with
- Method
  - Method is not adaptive, as it uses a fixed number of 3 iterations for the first frame, and 1 for others.
  - Not convincing why a transformer decoder block is required for preheating.
- Experiments
  - Baseline performance does not match original papers
  - evaluation metrics are not consistent across different datasets
  - sequence length generalisation
  - computational costs
  - Method was not tested on more challenging datasets like OVIS.
- Analysis
  - Insuffient comparisons to prior work (ie BO-QSA) and analysis into why the method works.
  - Discussions in the paper are not supported by sufficient evidence.

Whilst many of the experimental concerns were addressed during the rebuttal, others still remain (ie performance on OVIS is still underwhelming). The concerns relating to the methodology and analysis were not as well answered in the rebuttal. Moreover, the rebuttal would constitute substantial changes to the paper which would require another round of review. Therefore, the final decision is to reject the paper.

**Reviewer Concerns:**

Reviewer FgBa concerns were addressed.
Reviewer eLvL concerns were not sufficiently addressed.
Reviewer xdve concerns were partially addressed.
Refer to above for more details.

**Reviewer Scores:**

Reviewer FgBa likely to upgrade to weak accept.
Reviewer eLvL remain at reject.
Reviewer xdve remaining at weak reject.

---

### Decision · Program_Chairs · 2026-01-26

Reject